


**Assessing Climate Change-Induced Flood Risk in the**
**Conasauga River Watershed: An Application of Ensemble**
**Hydrodynamic Inundation Modeling**
Tigstu T. Dullo,[1] Sudershan Gangrade,[2,3] Mario Morales-Hernández,[3,4] Md Bulbul
Sharif,[5] Alfred J. Kalyanapu,[1,*] Shih-Chieh Kao,[2,3] Sheikh Ghafoor,[5] and Moetasim
Ashfaq [3,4]
[1] Department of Civil and Environmental Engineering, Tennessee Technological
University, Cookeville, TN 38505, USA
[2] Environmental Sciences Division, Oak Ridge National Laboratory, Oak Ridge, TN
37831, USA
[3] Climate Change Science Institute, Oak Ridge National Laboratory, Oak Ridge, TN
37831, USA
[4] Computational Sciences and Engineering Division, Oak Ridge National Laboratory,
Oak Ridge, TN 37831, USA
[5] Department of Computer Science, Tennessee Technological University, Cookeville, TN
38505, USA
*Corresponding Author
Alfred J. Kalyanapu, PhD
1020 Stadium Drive, P O Box 5015
Cookeville, TN 38505
Telephone: 931-372-3561
Email Address: akalyanapu@tntech.edu

Notice: This manuscript has been authored by UT-Battelle, LLC, under contract DE-AC05-
00OR22725 with the US Department of Energy (DOE). The US government retains and the
publisher, by accepting the article for publication, acknowledges that the US government retains a
nonexclusive, paid-up, irrevocable, worldwide license to publish or reproduce the published form
of this manuscript, or allow others to do so, for US government purposes. DOE will provide
public access to these results of federally sponsored research in accordance with the DOE Public
Access Plan (http://energy.gov/downloads/doe-public-access-plan).





**Abstract**

This study evaluates the impact of potential future climate change on flood regimes,

floodplain protection, and electricity infrastructures across the Conasauga River
Watershed in the southeastern United States through ensemble hydrodynamic inundation
modeling. The ensemble streamflow scenarios were simulated by the Distributed
Hydrology Soil Vegetation Model (DHSVM) driven by (1) 1981–2012 Daymet
meteorological observations, and (2) eleven sets of downscaled global climate models
(GCMs) during the 1966–2005 historical and 2011–2050 future periods. Surface
inundation was simulated using a GPU-accelerated Two-dimensional Runoff Inundation
Toolkit for Operational Needs (TRITON) hydrodynamic model. Nine out of the eleven
GCMs exhibit an increase in the mean ensemble flood inundation areas. Moreover, at the
1% annual exceedance probability level, the flood inundation frequency curves indicate a
~16 km$^2$ increase in floodplain area. The assessment also shows that even after flood-
proofing, four of the substations could still be affected in the projected future period. The
increase in floodplain area and substation vulnerability highlights the need to account for
climate change in floodplain management. Overall, this study provides a proof-of-
concept demonstration of how the computationally intensive hydrodynamic inundation
modeling can be used to enhance flood frequency maps and vulnerability assessment
under the changing climatic conditions.

**Keywords:** Flood simulation; Climate change; Critical electricity infrastructure;
Floodplain protection standards.





## 1. Introduction


Floods are costly disasters that affect more people than any other natural hazard

around the world (UNISDR, 2015). Major factors that can exacerbate flood damage
include population growth, urbanization, and climate change (Birhanu et al., 2016;
Winsemius et al., 2016; Alfieri et al., 2017; Alfieri et al., 2018; Kefi et al., 2018). Recent
observations exhibit an increase in the frequency and the intensity of extreme
precipitation events (Pachauri and Meyer, 2014), which have strengthened the magnitude
and frequency of flooding (Milly et al., 2002; Langerwisch et al., 2013; Alfieri et al.,
2015a; Alfieri et al., 2018; Mora et al., 2018). As a result, the damage and cost of
flooding have substantially increased across the United States (US) (Pielke Jr. and
Downton, 2000; Pielke Jr. et al., 2002; Ntelekos et al., 2010; Wing et al., 2018) and the
rest of the world (Hirabayashi et al., 2013; Arnell and Gosling, 2014; Alfieri et al.,
2015b; Alfieri et al., 2017; Kefi et al., 2018).

Since 1968, the National Flood Insurance Program (NFIP), administered by the

Federal Emergency Management Agency (FEMA), has implemented floodplain
regulation standards in the US to mitigate the escalating flood losses (FEMA, 2002). For
communities participating in the NFIP, flood insurance is required for structures located
within the 1% annual exceedance probability (AEP) flood zone (i.e., areas with
probability of flooding ≥ 1% in any given year; FEMA, 2002). However, existing
floodplain protection standards have proven to be inadequate (Galloway et al., 2006;
Ntelekos et al., 2010; Tan, 2013; Blessing et al., 2017; HCFCD, 2018), and climate
change can likely exacerbate these issues (Olsen, 2006; Ntelekos et al., 2010; Kollat et
al., 2012; AECOM, 2013; Wobus et al., 2017; Nyaupane et al., 2018; Pralle, 2019). For



instance, the streamflow AEP thresholds and synthetic hydrographs used to simulate the
flood zones were derived purely based on historic observations that may underestimate
the intensified hydrologic extremes in the projected future climatic conditions. Although
the possible change of future streamflow AEP thresholds may be evaluated by an
ensemble of hydrologic model outputs driven by multiple downscaled and bias-corrected
climate models (e.g., Wobus et al., 2017), the extension from maximum streamflow to
maximum flood zone is not trivial, and cannot be explicitly addressed through the
conventional deterministic inundation modeling approach.

The increases in the magnitude and frequency of flooding, in addition to the

inadequacy of floodplain measures and the high costs of hardening (Wilbanks et al.,
2008; Farber-DeAnda et al., 2010; Gilstrap et al., 2015), have put electricity
infrastructures at risk (Zamuda et al., 2015; Zamuda and Lippert, 2016; Cronin et al.,
2018; Forzieri et al., 2018; Mikellidou et al., 2018; Allen-Dumas et al., 2019). In
particular, electricity infrastructures which lie in areas vulnerable to flooding can
experience floodwater damages that may lead to changes in their energy production and
consumption (Chandramowli and Felder, 2014; Ciscar and Dowling, 2014; Bollinger and
Dijkema, 2016; Gangrade et al., 2019). For instance, flooding can rust metals, destroy
insulation, and damage interruption capacity (Farber-DeAnda et al., 2010; Vale, 2014;
NERC, 2018; Bragatto et al., 2019). It is estimated that nearly 300 energy facilities are
located on low-lying lands vulnerable to sea-level rise and flooding in the lower 48 US
states, (Strauss and Ziemlinski, 2012).

Several studies have assessed the vulnerability of electricity infrastructures to

flooding (Reed et al., 2009; Winkler et al., 2010; Bollinger and Dijkema, 2016; Fu et al.,


2017; Pant et al., 2017; Bragatto et al., 2019; Gangrade et al., 2019). Although some of
these studies focused on evaluating the resilience of electricity infrastructures against
flood hazard and/or climate change, only a few of them evaluated site-specific inundation
risk and quantified impacts of climate change-induced flooding on electricity
infrastructures under different future climate scenarios. Again, one main challenge is
associated with the high computational costs to effectively transform ensemble
streamflow projections into ensemble surface inundation projections through
hydrodynamic models. With the enhanced inundation models and high performance
computing (HPC) capabilities (Morales-Hernández et al., 2020a), this challenge can be
gradually overcome for more spatially explicit flood vulnerability assessment.
The objective of this study is to demonstrate the applicability of a computationally
intensive ensemble inundation modeling approach to better understand how climate
change may affect flood regimes, floodplain regulation standards, and the vulnerability of
existing infrastructures. The unique aspects of this study are the application of an
integrated climate-hydrologic-hydraulic modeling framework for:
(1) Evaluating the changes in flood regime using high-resolution ensemble flood

inundation maps. The ensemble-based approach is able to incorporate the large

hydrologic interannual variability and model uncertainty that cannot be captured

through the conventional deterministic flood map.

(2) Enabling direct frequency analysis of ensemble flood inundation maps that

correspond to historic and projected future climate conditions. This approach

provides an alternative floodplain delineation technique to the conventional




approach, in which a single deterministic design flood value is used to develop a
flood map with a given exceedance probability.
(3) Evaluating the vulnerability of electricity infrastructures to climate change-
induced flooding and assessing the adequacy of existing flood protection
measures using ensemble flood inundation. This information will help floodplain
managers to identify the most vulnerable infrastructures and recommend suitable
adaptation measures.
The following technique was adopted in this study. First, we generated streamflow
projection by utilizing an ensemble of simulated streamflow hydrographs driven by both
historical observations and downscaled climate projections (Gangrade et al., 2020) as
inputs for hydrodynamic inundation modeling as presented in section 2.2. Then, we set
up and calibrated a 2D hydrodynamic inundation model, Two-dimensional Runoff
Inundation Toolkit for Operational Needs (TRITON; Morales-Hernández et al., 2020b),
in our study area which is presented in section 2.3. For inundation modeling, sensitivity
analyses were conducted on three selected parameters to quantify and compare their
respective influences on modeled flood depths and extents. The performance of TRITON
was then evaluated by comparing a simulated 1% AEP flood map with the reference 1%
AEP flood map from the Federal Emergency Management Agency (FEMA). Finally, as
presented in sections 2.4 and 2.5, ensemble inundation modeling was performed to
develop flood inundation frequency curves and maps, and to assess the vulnerability of
electricity infrastructures under a changing climate, respectively.
The article is organized as follows: the data and methods are discussed in Section 2;
Section 3 presents the result and discussion; and the summary is presented in Section 4.




## 2. Data and Methods

### 2.1. Study Area

Our study area is the Conasauga River Watershed (CRW) located in southeastern Tennessee and northwestern Georgia (Figure 1). The CRW is an eight-digit Hydrologic Unit Code (HUC08) subbasin (03150101) with a total drainage area of ~1880 km$^2$. The northeastern portions of the watershed are rugged, mountainous areas largely covered with forests (Ivey and Evans, 2000; Elliott and Vose, 2005). The CRW, which is one headwater basin of the Alabama-Coosa-Tallapoosa (ACT) River Basin, rises high on the Blue Ridge Mountains of Georgia and Tennessee and flows for 145 km before joining the Coosawattee River to form the Oostanaula River (Ivey and Evans, 2000; USACE, 2013). The CRW climate is characterized by warm, humid summers, and mild winters with mean annual temperature of 15 to 20 °C and average annual precipitation of 1300 to 1400 mm (FIS, 2007; FIS, 2010; Baechler et al., 2015). The watershed encompasses four counties: Bradley, Polk, Fannin, Murray, and Whitfield. It also includes the cities of Dalton and Chatsworth, Georgia. There is no major reservoir located in the CRW.

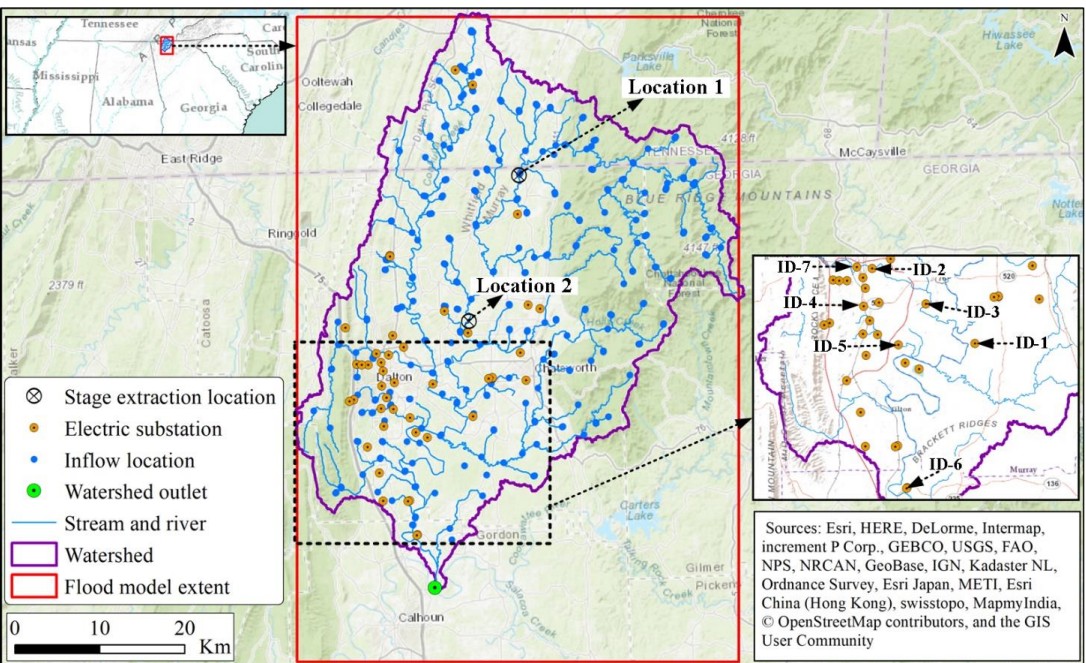


Figure 1. Conasauga River Watershed study area location, model extent, electric
substations, and inflow locations. Background layer source: © OpenStreetMap
contributors 2020. Distributed under a Creative Commons BY-SA License.

**2.2.  Streamflow Projections**
The ensemble streamflow projections were generated by a hierarchical modeling
framework, which started with regional climate downscaling followed by hydrologic
modeling (Gangrade et al., 2020). The climate projections were generated by dynamically
downscaling of 11 GCMs from the Coupled Model Intercomparison Project Phase-5
(CMIP5) data archive. Each GCM was used as lateral and lower boundary forcing in a
regional climate model RegCM4 (Giorgi et al., 2012) at a horizontal grid spacing of 18
km over a domain that covered continental US and parts of Canada and Mexico (Ashfaq
et al., 2016) (Table 1). Each RegCM4 integration covered 40 years in the historic period




(1966–2005; hereafter baseline) and another 40 years in the future period (2011–2050)
under Representative Concentration Pathway 8.5 (RCP 8.5) emission scenario, with a
combined 880 years of data across all RegCM4 simulations.

Table 1. Summary of the 11 dynamically downscaled climate models (adopted from
Ashfaq et al., 2016).

| S. No. | Climate model name | Number of flood events per climate model | Time period | |
|---|---|---|---|---|
| 1 | ACCESS1-0 | | | |
| 2 | BCC-CSM1-1 | | | |
| 3 | CCSM4 | | | |
| 4 | CMCC-CM | | | |
| 5 | FGOALS-g2 | | 1966–2005 | 2011-2050 |
| 6 | GFDL-ESM2M | 40 | (Baseline) | (Future/RCP |
| 7 | MIROC5 | | | 8.5) |
| 8 | MPI-ESM-MR | | | |
| 9 | MRI-CGCM3 | | | |
| 10 | NorESM1-M | | | |
| 11 | IPSL-CM5A-LR | | | |


The RegCM4 simulated daily precipitation and temperature were further statistically
bias-corrected to a spatial resolution of 4 km following a quantile mapping technique,
described in Ashfaq et al. (2010, 2013). The 4 km Parameter-elevation Regressions on
Independent Slopes Model (PRISM; Daly et al., 2008) data was used as the historic
observations to support bias-correction. In the baseline period, the simulated quantiles of
precipitation and temperature were corrected by mapping them onto the observed



quantiles. In the future period, the monthly quantile shifts were calculated based on the
simulated baseline and future quantiles which were subsequently added to the bias
corrected baseline quantiles to generate bias-corrected monthly future data. Finally, the
monthly bias-corrections were distributed to the daily values while preserving in each
time period. This approach substantially improves the biases in the modeled daily
precipitation and temperature while preserving the simulated climate change signal.
Further details of the bias-correction are provided in Ashfaq et al. (2010, 2013) while the
information regarding the RegCM4 configuration, evaluation and future climate
projections are detailed in Ashfaq et al. (2016).

The hydrologic simulations were then conducted using the Distributed Hydrology

Soil Vegetation Model (DHSVM; Wigmosta et al., 1994), which is a process-based high-
resolution hydrologic model that can capture heterogeneous watershed processes and
meteorology at a fine resolution. DHSVM uses spatially distributed parameters, including
topography, soil types, soil depths, and vegetation types. The input meteorological data
includes precipitation, incoming shortwave and longwave radiation, relative humidity, air
temperature and wind speed (Wigmosta et al., 1994; Storck et al., 1998; Wigmosta et al.,
2002). The DHSVM performance and applicability has been reported in various earlier
climate and flood related studies (Elsner et al., 2010; Hou et al., 2019; Gangrade et al.,
2018, 2019, 2020). A calibrated DHSVM implementation from Gangrade et al. (2018) at
90 m grid spacing was used to produce 3-hourly streamflow projections using the
RegCM4 meteorological forcings described in the previous section (Table 1). In addition,
a control simulation driven by 1981–2012 Daymet meteorologic forcings (Thornton et
al., 1997) was conducted for model evaluation and validation. The hydrologic simulations



used in this study are a part of a larger hydroclimate assessment effort for the ACT River
Basin, as detailed in Gangrade et al. (2020). Since there is no major reservoir in the
CRW, the additional reservoir operation module (Zhao et al., 2016) was not needed in
this study.

Note that while the ensemble streamflow projections based on dynamical

downscaling and high-resolution hydrologic modeling from Gangrade et al. (2020) are
suitable to explore extreme hydrologic events in this study, they do not represent the full
range of possible future scenarios. Additional factors such as other GCMs, RCP
scenarios, downscaling approaches, and hydrologic models and parameterization may
also affect future streamflow projections. In other words, although these ensemble
streamflow projections can tell us how likely the future streamflow magnitude may
change from the baseline level, they are not the absolute prediction into the future. In
practice, these modeling choices will likely be study-specific based on the agreement
among key stakeholders. It is also noted that the new Coupled Model Intercomparison
Project Phase-6 (CMIP6) data have also become available to update the ensemble
streamflow projections, but is not pursued in this study.
**2.3.  Inundation Modeling**

The ensemble inundation modeling was performed using TRITON, which is a

computationally enhanced version of Flood2D-GPU (Kalyanapu et al., 2011). TRITON
allows parallel computing using multiple graphics processing units (GPUs) through a
hybrid Message Passing Interface (MPI) and Compute Unified Device Architecture
(CUDA) (Morales-Hernández et al., 2020b). TRITON solves the nonlinear hyperbolic
shallow water equations using an explicit upwind finite-volume scheme, based on Roe's





linearization. The shallow water equations are a simplified version of the Navier-Stokes
equations in which the horizontal momentum and continuity equations are integrated in
the vertical direction (see Morales-Hernández et al., (2020b), for further model details).
An evaluation of TRITON performance for the CRW is presented and discussed in
Section 3.3.

TRITON's input data includes digital elevation model (DEM), surface roughness,

initial depths, flow hydrographs, and inflow source locations (Kalyanapu et al., 2011;
Marshall et al., 2018; Morales-Hernández et al., 2020a; Morales-Hernández et al.,
2020b). In this study, the hydraulic and geometric parameters from the flood model
evaluation section (Section 3.3) were used in the flood simulation. The topography was
represented using the one-third arc-second (~10 m) spatial resolution DEM (Archuleta et
al., 2017) from the US Geological Survey (USGS). To improve the quality of the base
DEM, as discussed in the flood model evaluation section, the main channel elevation was
reduced by 0.15 m. Elevated roads and bridges that obstruct the flow of water were also
removed. For surface roughness, we used a single channel Manning's n value of 0.05 and
a single floodplain Manning's n value of 0.35. The selection of channel and floodplain
Manning's n value was based on the Whitfield County Flood Insurance Study (FIS,
2007), which reported a range of Manning's n values estimated from field observations
and engineering judgment for about 15 streams inside the CRW (section 3.2).
Furthermore, a water depth value of 0.35 m was defined for the main river channel as an
initial boundary condition. The zero velocity gradients were used as the downstream
boundary condition. Further discussion of model parameter sensitivity and model
evaluation are provided in sections 3.2 and 3.3.





The simulated DHSVM streamflow was used to prepare inflow hydrographs for
ensemble inundation modeling. To provide a large sample size for frequency analysis, we
selected all annual maximum peak streamflow events (the maximum corresponded to the
outlet of CRW [Figure 1]) from the 1981–2012 control simulation (32 years), the 1966–
2005 baseline simulation (440 years; 40 years × 11 models), and the 2011–2050 future
simulation (440 years; 40 years × 11 models), with a total of 912 events. For each annual
maximum event, the 3-hour timestep, 10-day hydrographs (which capture the peak CRW
outlet discharge) across all DHSVM river segments were summarized. Following a
procedure similar to Gangrade et al. (2019), these streamflow hydrographs were
converted to TRITON inputs at 300 inflow locations selected along the NHD+ river
network in the CRW (Figure 1). The TRITON model extent, shown in Figure 1, has an
approximate area of 3945 km$^2$ and includes ~44 million model grid cells (7976 rows ×
5474 columns in a uniform structured mesh). The ensemble flood simulations resulted in
gridded flood depth and velocity output at 30-minute intervals. The simulations generated
an approximately 400 Terabyte data and utilized ~2000 node hours on the Summit
supercomputer, managed by the Oak Ridge Leadership Computing Facility at Oak Ridge
National Laboratory.
**2.4.   Flood Inundation Frequency Analysis**
Given the nature of GCM experiments, each set of climate projections can be
considered as a physics-based realization of historic and future climate under specified
emission scenarios. Therefore, an ensemble of multimodel simulations can effectively
increase the data lengths and sample sizes that are keys to support frequency analysis,
especially for low-AEP events. In this study, we conducted flood frequency analyses



separately for the 1966–2005 baseline and 2011–2050 future periods so that the
difference between the two periods represent the changes in flood risk due to climate
change.

To prepare the flood frequency analysis, we first calculated the maximum flood depth

at every grid in each simulation. A minimum threshold of 10 cm flood depth was used to
judge whether a cell was wet or dry (Gangrade et al., 2019). Further, for a given grid cell,
if the total number of non-zero flood depth values (i.e., of the 440 depth values) was less
than 30, the grid cell was also considered dry. This threshold was selected based on the
minimum sample size requirement for flood depth frequency analysis suggested by Li et
al. (2018). Next, we calculated the maximum flooded area (hereafter used alternatively
with "floodplain area") for each simulation. A log-Pearson Type III (LP3) distribution
was then used for frequency analysis following the guidelines outlined in Bulletins 17B
(USGS, 1982; Burkey, 2009) and 17C (England Jr. et al., 2019). Two types of LP3 fitting
were performed. The first type of fitting is event-based that fitted LP3 on the maximum
inundation area across all ensemble members. The second type of fitting is grid-based
(more computationally intensive) that fitted LP3 on the maximum flood depth at each
grid cell across all ensemble members. For both types of fittings, the frequency estimates
at 4%, 2%, 1%, and 0.5% AEP (corresponding to 25-, 50-, 100-, and 200-year return
levels) were derived for further analysis.

It is also noted that in addition to the annual maximum event approach used in this

study, one may also use the peak-over-threshold (POT) approach which can select
multiple streamflow events in a very wet year. While such an approach can lead to higher
extreme streamflow and inundation estimates, the timing of POT samples is fully





governed by the occurrences of wet years. In other words, if the trend of extreme
streamflow is significant in the future period, the POT samples will likely occur more in
the far future period. We hence select the annual maximum event approach that can
sample maximum streamflow events more evenly in time, which can better capture the
evolution of extreme events with time under the influence of climate change.
**2.5. Vulnerability of Electricity Infrastructure**
The vulnerability of electricity infrastructures to climate change-induced flooding
was evaluated using the ensemble flood inundation results. The 44 electric substations
(Figure 1) collected from the publicly available Homeland Infrastructure Foundation-
Level Data (HIFLD, 2019) were considered to be the electrical components susceptible to
flooding. To evaluate the vulnerability of these substations, we overlapped the maximum
flood extent from each ensemble member with all substations to identify the substations
that might be inundated under the baseline and future climate conditions. Further, as an
additional flood hazard indicator, the duration of inundation was estimated at each of the
affected substations using the ensemble flood simulation results.
The vulnerability analysis was performed for two different flood mitigation scenarios.
In the first scenario, we assumed that no flood protection measures were provided at all
substations. Hence, the substations that intersected with the flood footprint were
considered to be failed. In the second scenario, it was assumed that flood protection
measures were adopted for all substations following the FEMA P-1019 recommendation
(FEMA, 2014). According to FEMA P-1019 (FEMA, 2014), for emergency power
systems within critical facilities, the highest elevation among (1) the base flood elevation
(BFE: 1% FEMA AEP flood elevation) plus 3 feet (~0.91 m), (2) the locally adopted



design flood elevation, and (3) the 500-year flood elevation can be used to design flood
protection measures. Since the three recommended elevations were not available at all
substation locations, we focused only on the BFE plus ~0.91 m option. In addition, since
in the CRW the majority of existing flood insurance maps were classified as Zone A—
meaning that the special flood hazard areas were determined by approximate methods
without BFE values (FEMA, 2002)—we used the maximum flood depth values across all
control simulation years as the BFE values in this second mitigation scenario.

During the vulnerability analysis, we also assumed that (1) the one-third arc-second

spatial resolution DEM might reasonably represent the elevation of substations, (2)
existing substations would remain functional and would not be relocated, and (3) no
additional hardening measures (i.e., protections such as levees, berms, anchors, and
housings) will be adopted in the future period. Also, the cascading failure of a substation
due to grid interconnection was not considered in this study.
**3.   Results and Discussion**
**3.1.   Streamflow Projections**

This section presents a comparison of the annual maximum peak streamflow (at the

outlet of CRW) used in the control, baseline, and future simulations. The sample size
included 32 events from the control (1981–2012) simulation, 440 events from the
baseline (1966–2005) simulations, and another 440 events from the future (2011–2050)
simulations. These samples are illustrated in box and whisker plots in Figure 2, where
central mark indicate the median, while bottom and top edges indicate the $25^{th}$ and $75^{th}$
percentiles respectively. The whiskers extend to the furthest data points not considered
outliers, which correspond to approximately $\pm$ 2.7 standard deviations and 99.3%





coverage if the data are normally distributed. As is evident from Figure 2, the
distributions of annual maximum peak streamflow values in the control and baseline
simulations are comparable. The upper and lower whiskers in the control simulation are
727.6 $m^3/s$ and 84.2 $m^3/s$, which compare well to the 722.5 $m^3/s$ and 65.2 $m^3/s$ values in
the baseline simulation. A larger number of outliers are present in the baseline
simulation, which is due to the larger sample size (440 versus 32). Under the future
projection, an increase in the maximum peak streamflow is shown, where the upper
whisker in the future projection is ~21% higher than the baseline. Moreover, the
maximum of distribution in the future climate (2036.7 $m^3/s$) is also much higher than that
in the baseline climate (1436.7 $m^3/s$), suggesting a higher future flood risk in the CRW.
The increasing trend of streamflow extremes in the CRW is consistent with the overall
findings in the ACT River Basin (Gangrade et al., 2020).

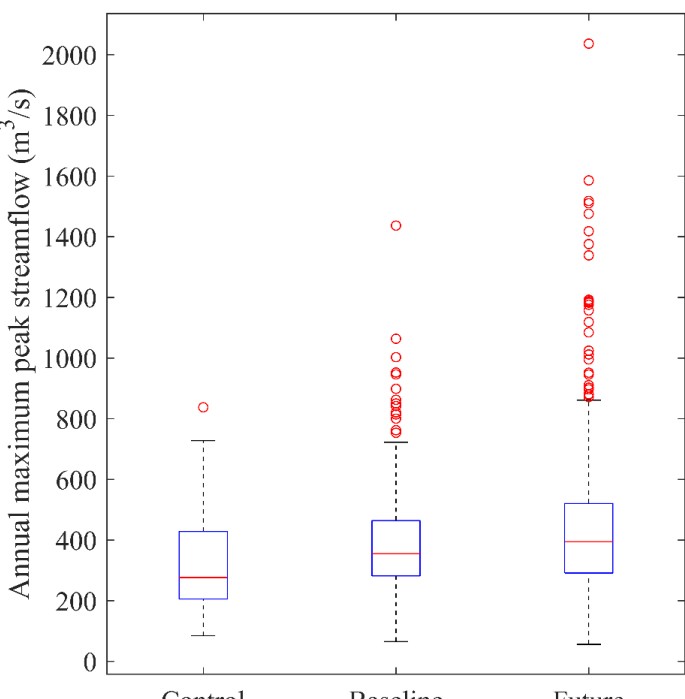


Figure 2. A comparison of annual maximum peak streamflow at the outlet of Conasauga
River Watershed. The sample size includes 32 events from the control (1981–2012), 440
from the baseline (1966–2005), and another 440 from the future (2011–2050) periods.
**3.2. Sensitivity Analysis for Flood Model**

For a better understanding and selection of suitable TRITON parameters, a series of

sensitivity analyses were conducted using different combinations of Manning's
roughness, initial water depths, and river bathymetry correction factors (Table 2).





Table 2. Summary of hydraulic and geometric parameters used in the sensitivity analysis.

| Sensitivity parameter | Scenario | Initial water depth values (m) | Surface roughness (Manning's n values) | Bathymetry correction factor (m) |
|---|---|---|---|---|
| Initial water depth | 1 | 0.00 | $n_{ch}$ =0.050 / $n_{fldpl}$ =0.350 | -0.15 |
| | 2 | 0.15 | | |
| | 3 | 0.35 | | |
| | 4 | 0.45 | | |
| | 5 | 0.55 | | |
| | 6 | 0.65 | | |
| Surface roughness | 1 | 0.35 | N_1: $n_{ch}$ =0.035 / $n_{fldpl}$ =0.06 | -0.15 |
| | 2 | | N_2: $n_{ch}$ =0.040 / $n_{fldpl}$ =0.25 | |
| | 3 | | N_3: $n_{ch}$ =0.045 / $n_{fldpl}$ =0.30 | |
| | 4 | | N_4: $n_{ch}$ =0.050 / $n_{fldpl}$ =0.35 | |
| | 5 | | N_5: $n_{ch}$ =0.055 / $n_{fldpl}$ =0.45 | |
| | 6 | | N_6: $n_{ch}$ =0.060 / $n_{fldpl}$ =0.50 | |
| Bathymetry correction factor | 1 | 0.35 | $n_{ch}$ =0.050 / $n_{fldpl}$ =0.350 | 0.00 |
| | 2 | | | -0.15 |
| | 3 | | | -0.45 |
| | 4 | | | -0.75 |
| | 5 | | | -1.00 |
| | 6 | | | -1.25 |

Note: $n_{ch}$ represents the Manning's n value in the main channel and $n_{fldpl}$ represents the
Manning's n value in the floodplain areas.

In calibrating a hydraulic model, it is a common practice to adjust the estimated
Manning's n value, as it is the most uncertain and variable input hydraulic parameter
(Brunner et al., 2016). In this study, we tested six different scenarios (Table 2) based on
the Whitfield County Flood Insurance Study (FIS, 2007), which reported a range of
Manning's n values estimated from field observations and engineering judgment for
about 15 streams inside the CRW. To establish an initial condition for TRITON, a
sensitivity analysis was performed on selected initial water depth values (ranging from
0 m to 0.65 m, Table 2) to understand their relative effects. To select ranges for the initial
water depth, we summarized the observed water depth values that corresponds to low





flow values at five USGS gauge stations inside the CRW. The distribution of observed
water depth values from the five gauges showed average values ranging from 0.25 to
0.65m. Existing DEM products, even those with high spatial resolution (i.e., 10 m or
finer), do not represent the elevation of river bathymetry accurately (Bhuyian et al.,
2014). For the CRW, Bhuyian et al. (2019) found that the one-third arc-second spatial
resolution base DEM over-predicted the inundation extent because of the bathymetric
error, which reduced the channel conveyance. In this study, we tested various bathymetry
correction factors (ranging from −1.25 m to 0 m, Table 2) by reducing the DEM elevation
along the main channel to understand the sensitivity of TRITON.

The sensitivity analysis was performed using the February 13–22, 1990 flood event

that has the maximum discharge among all 32 control simulation events. To evaluate
relative sensitivity of TRITON, we extracted simulated flood depths at two arbitrary
selected locations (Figure 1) and estimated the relative inundation area differences. The
impacts of initial water depths were significant only at the beginning where low flow
values dominated the hydrographs (Figure 3a, 3d). Larger initial water depth values
generated higher flood inundation depths for both sample locations. Although the
differences in flood inundation extents relative to the dry bed show an increasing trend,
the relative differences are less than 1.4% (Figure 4a). Increase in the channel and
floodplain Manning's n values resulted in higher flood depths for both sample locations
(Figure 3b and 3e). The relative flood inundation area differences increase from about
23% to 31% (Figure 4b) when the channel and floodplain Manning's n values are
increased from 0.035 to 0.06 and from 0.06 to 0. 50, respectively. Reduction in the
elevation of river bathymetry (to improve the quality of the base DEM) results in a direct


increase in maximum flood depth due to change in the river conveyance (Figure 3c and
3f). It also results in a decrease in the maximum flood extent (Figure 4c), as more water
is allowed to transport through the main channel instead of the floodplain. Overall, the
results showed that TRITON was more sensitive to the Manning's n values than the
initial water depths and bathymetric correction factors.


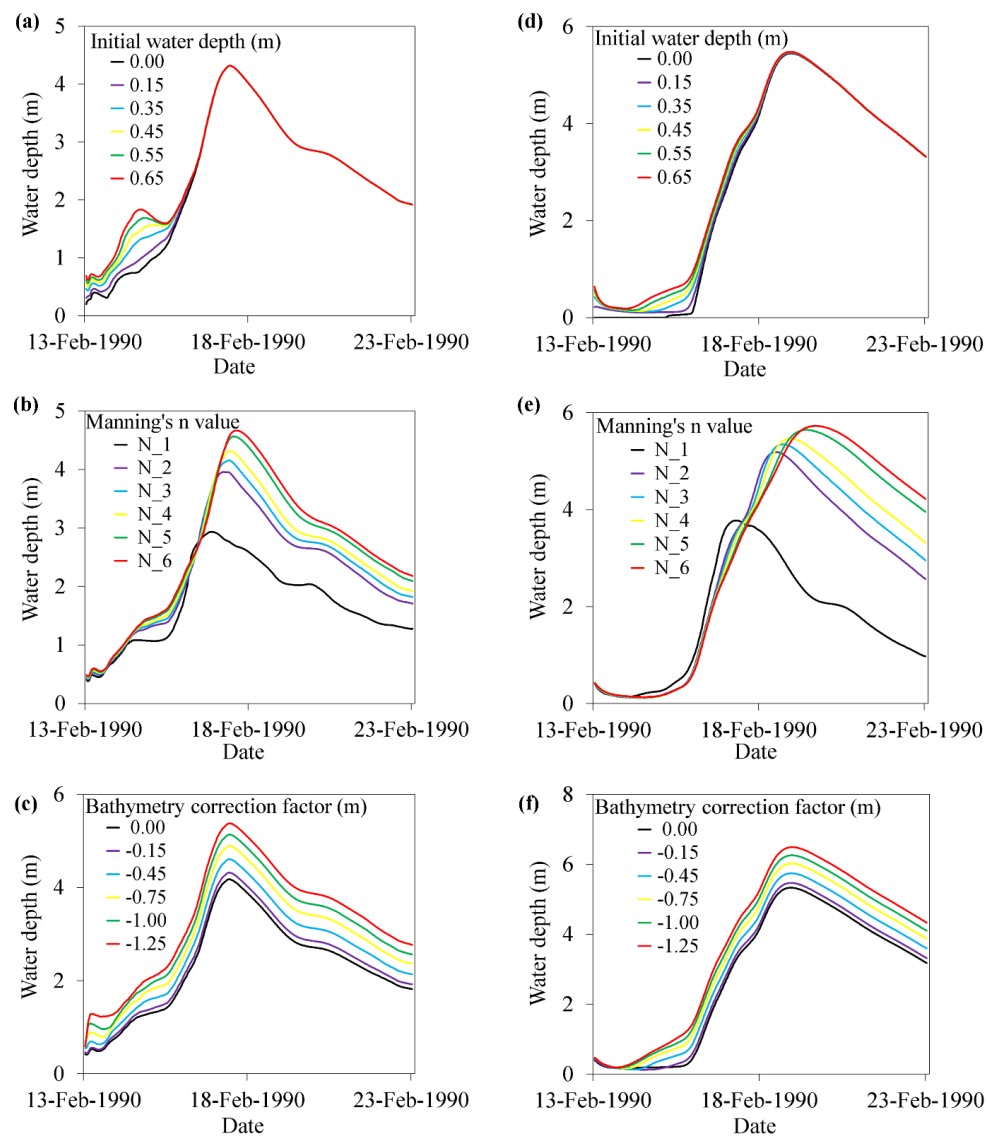


Figure 3. Simulated flood inundation depths extracted at location 1 (a, b, c) and at

location 2 (d, e, f). Note: Location 1 and 2 are shown in Figure 1. A description of the

Manning's n values (N_1 to N_6) can be found in Table 2.


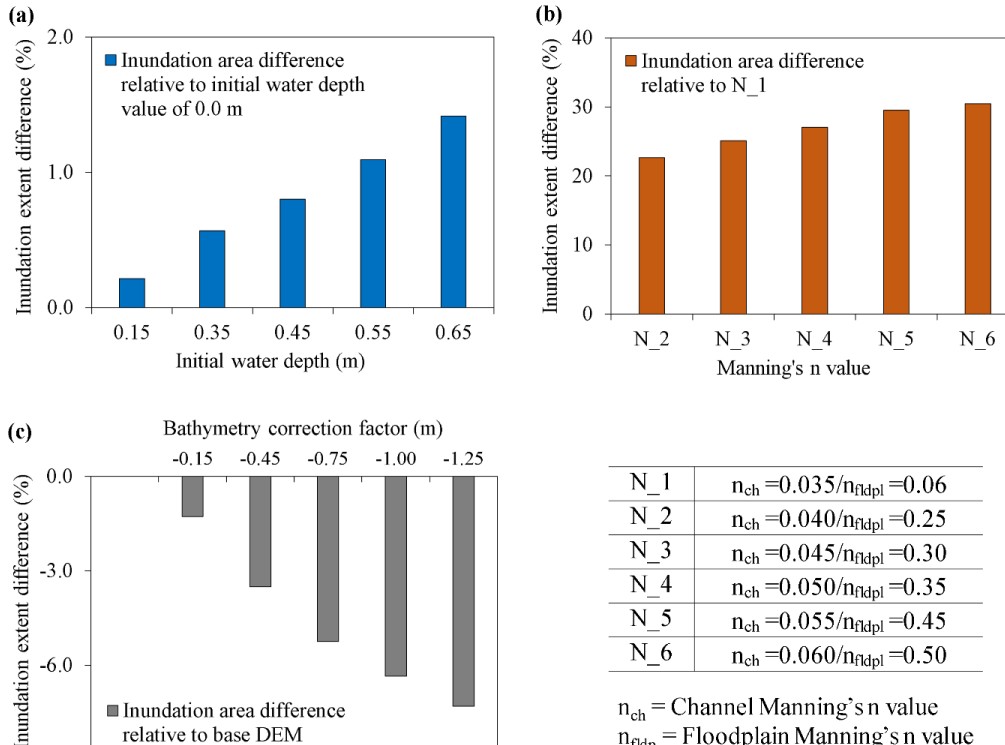

Figure 4. Change in simulated maximum flood inundation extents for (a) initial water

depth, (b) Manning's n value, and (c) bathymetry correction factor.

### 3.3. Flood Model Evaluation

Because of a lack of observed streamflow data in the CRW, the performance of

TRITON was evaluated by comparing the simulated 1% AEP flood map with the

published 1% AEP flood map from FEMA (FEMA, 2019). The purpose of this

assessment is to understand whether TRITON can provide comparable results to the

widely accepted FEMA flood estimates. While the FEMA AEP flood maps do not

necessarily represent complete ground truth, such a comparison is the best option given

the data challenge. Similar approach has been utilized by several previous studies in the


evaluation of large- scale flood inundation evaluation (Alfieri et al., 2014; Wing et al.,
2017; Zheng et al., 2018; Gangrade et al., 2019).
To derive the 1% AEP flood map using TRITON, the ensemble-based approach used
by Gangrade et al. (2019) was followed. The assessment started by preparing the
streamflow hydrographs used to construct the 1% AEP flood map. The 1981–2012
annual maximum peak events and their corresponding 10-day streamflow hydrographs
were extracted from the control simulation. These streamflow hydrographs were then
proportionally rescaled to match the 1% AEP peak discharge estimated at the watershed
outlet (Figure 1), following the frequency analysis procedures outlined in Bulletin 17C
(England Jr. et al., 2019). The streamflow hydrographs from control simulations were
used for the peak discharge frequency analysis.
The results reported in the sensitivity analysis were also used to help identify suitable
TRITON parameters. In addition to streamflow hydrographs, TRITON requires DEM,
initial water depth, and Manning's n value. To minimize the effect of bathymetric error in
the base DEM (Bhuyian et al., 2014; Bhuyian et al., 2019), we reduced the elevation
along the main channel by 0.15 m (i.e., a bathymetry correction factor). Although this
simple approach is unlikely to adjust the channel bathymetry to its true values, it can
improve the channel conveyance volume that is lost in the base DEM. To further improve
the quality of the base DEM, we removed elevated roads and bridges that could obstruct
the flow of water in some of the streams and rivers. An initial water depth of 0.35 m was
also selected in this study. For the surface roughness, a couple of flood simulations were
performed by adjusting the Manning's n values for the main channel and floodplain to
achieve satisfactory agreement between the simulated and the reference FEMA flood





map. We eventually selected a single channel Manning's n value of 0.05 and a single
floodplain Manning's n value of 0.35.
Three evaluation metrics, including fit, omission, and commission (Kalyanapu et al.,
2011) were used to quantify the differences between the modeled and reference flood
map. The measure of fit determines the degree of relationship, while the omission and
commission statistically compare the simulated and reference FEMA flood maps
(Kalyanapu et al., 2011). The comparison between the simulated maximum inundation
and the corresponding 1% AEP FEMA flood map showed 80.65% fit, 5.52%
commission, and 15.36% omission (Figure 5), demonstrating that the TRITON could
reasonably estimate flood inundation extent, depths, and velocities in the CRW. The
computational efficiency of TRITON can further support ensemble inundation modeling
to provide additional variability information that cannot be provided by the conventional
deterministic flood map.


Figure 5. Comparison of simulated maximum flood extent with the corresponding FEMA

1% AEP flood map for the Conasauga River Watershed. Background layer source: ©

OpenStreetMap contributors 2020. Distributed under a Creative Commons BY-SA

License.


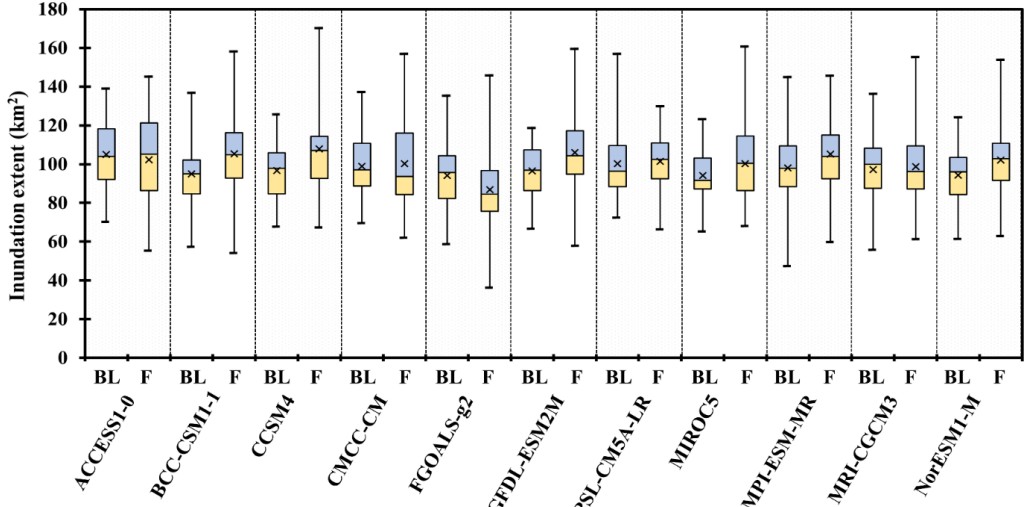

Figure 6. A summary of simulated maximum flood inundation extents obtained from the

baseline and future scenarios. The mean flooded area values are shown by × symbols.

Note: The suffix "_BL" represents baseline scenarios and the suffix "_F" represents

future scenarios.

## 3.5. Flood Inundation Frequency Curve and Map

Figure 7 shows the relationship between the 440 flooded area values (across 11

downscaled GCMs) and their corresponding peak streamflow at the watershed outlet, for

both the baseline and future periods. Overall, both results (Figure 7a and 7b) exhibit

strong nonlinear relationships with high $R^2$ values. The results suggest that peak

streamflow is a significant variable controlling the total flooded area, but the variability

of flooded area could not be explained by peak streamflow alone. For instance, in the

baseline period, the peak streamflow values of 423.63 m³/sec and 424.25 m³/sec

correspond to 106.85 km² and 94.89 km² floodplain areas, respectively (Figure 7a).




Similarly, in the future period, the peak streamflow values of 433.27 m³/sec and 434.21
m³/sec correspond to 110.76 km² and 99.26 km² floodplain areas (Figure 7b).

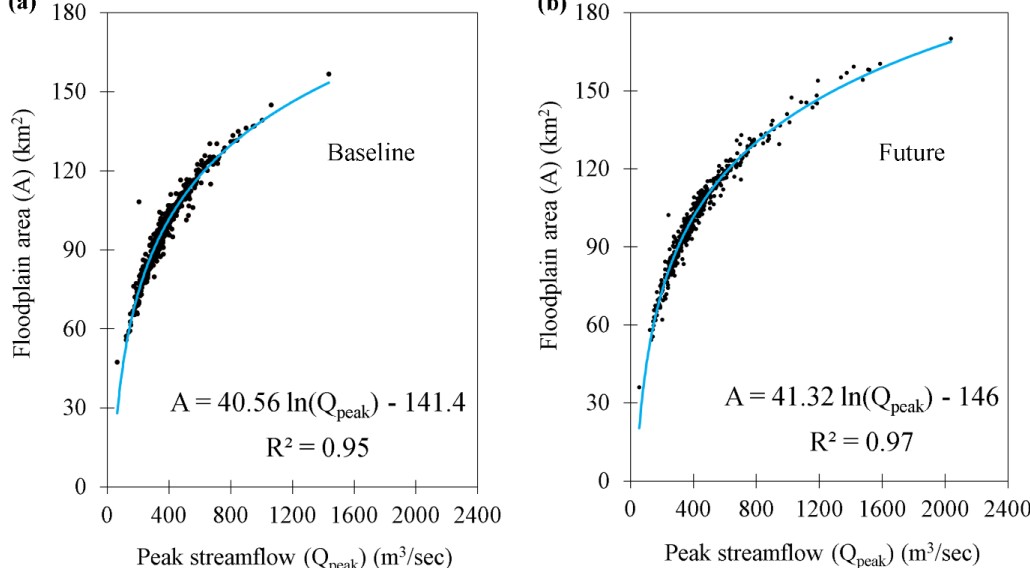


Figure 7. Relationship between floodplain areas and peak streamflow values at the
watershed outlet for (a) baseline and (b) future scenarios. The blue lines indicate the
logarithmic best-fit.

Figure 8 shows the event-based flood inundation frequency curves and their

corresponding 95% confidence intervals in both the baseline and future periods, for
which each frequency curve was derived using an ensemble of 440 years of data. The use
of long-term data helped reduce the uncertainty and add more confidence in the
evaluation of the lower AEP estimates. This type of assessment cannot be achieved using
only historic streamflow observations, for which the limited records present a major





challenge for lower AEP estimates. For most of the exceedance probabilities, the flooded
areas projected an increase in the inundation areas in the future period when compared to
the baseline period. The 1% AEP flood shows an ~16 km$^2$ increase in the inundation area
(137.75 km$^2$ in the baseline period versus 153.43 km$^2$ in the future period) (Figure 8).
Similar results can be observed in inundation frequency curves developed for other AEPs
(not shown).

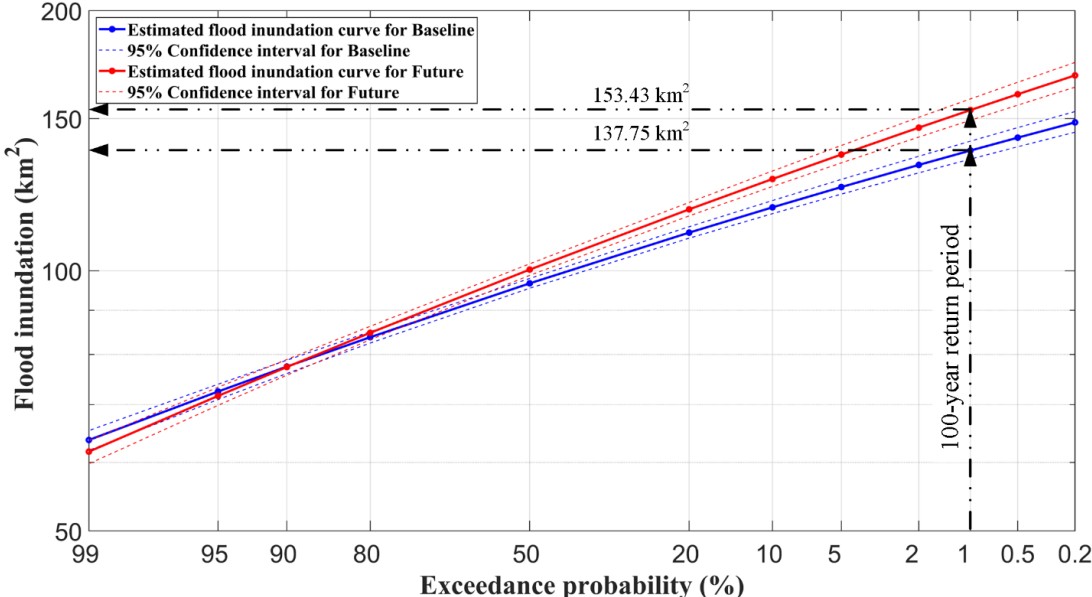


Figure 8. A summary of flood inundation frequency curves for the baseline and future
periods.

The grid-based flood depth frequency results at 0.5%, 1%, 2%, and 4% AEP levels

are illustrated in Figure 9. In each panel, the projected change (i.e., future minus baseline)
at each grid is shown. The corresponding histogram across the entire study area is


presented in Figure 10. Based on these comparisons, it is estimated that the flood depth
values at ~80% of grid cells would increase by 0.2 to 1.5 m due to projected changes in
climate (Figure 10). For 0.5% and 1% AEP flood depth frequency maps (Figure 9a and
9b), the changes in flood depth were more pronounced in the lower part of the CRW, near
the City of Dalton (where there are large population settlements), thereby increasing the
likelihood of population exposure to flood risk in the future period. Furthermore, for the
1% flood depth frequency map (Figure 9b), the projected increase in flood depths and
spatial extent has the potential to extend the flood damage far beyond the FEMA's
current base floodplain area. Therefore, these results highlight the need for climate
change consideration in the floodplain mapping. The approach presented in this study can
provide an alternative floodplain delineation technique, as it can be applied to develop
flood depth frequency maps that are reflective of the future climate.


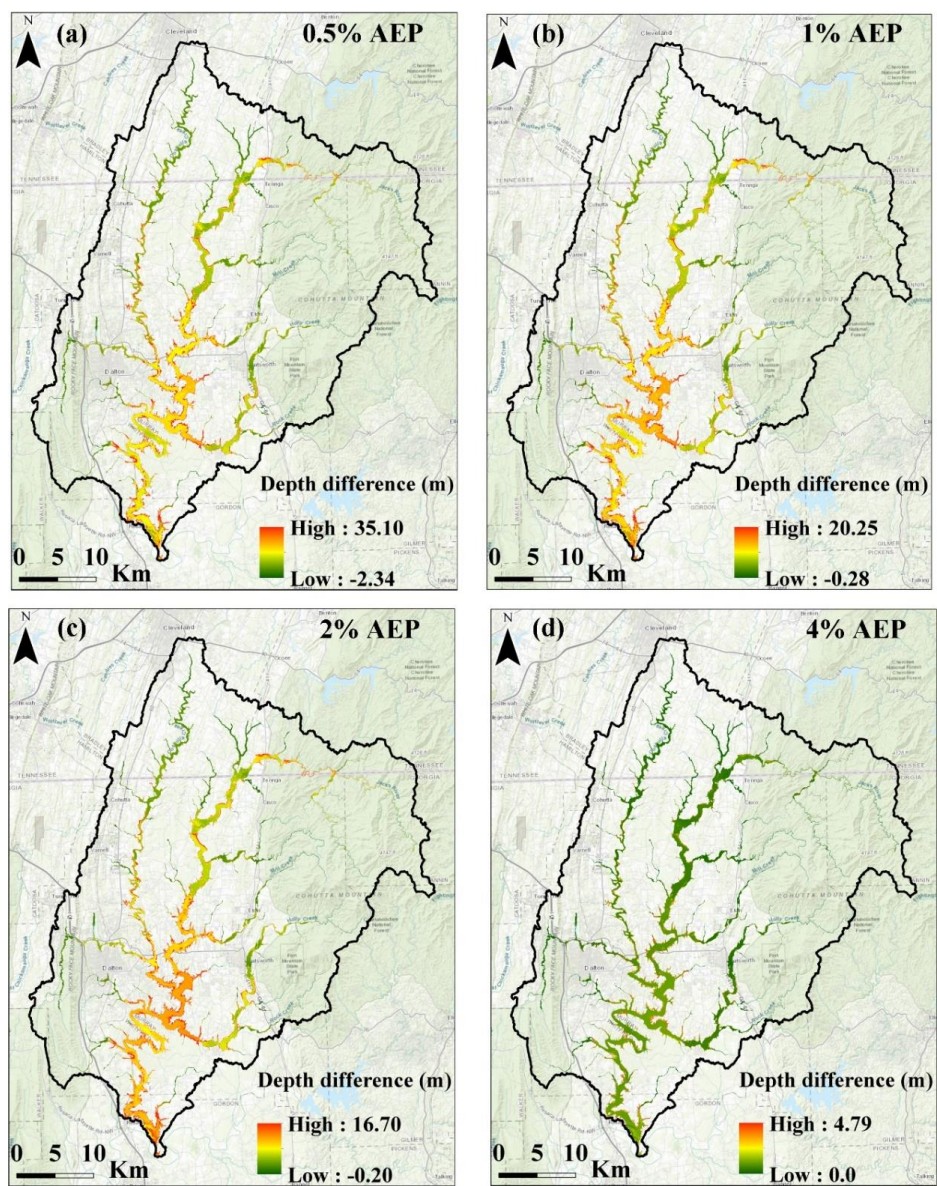

Figure 9. Projected change (future minus baseline period) in flood depth frequency maps
for (a) 0.5%, (b) 1%, (c) 2%, and (d) 4% AEPs. ArcGIS background layer sources: ESRI,
HERE, Garmin, Intermap, GEBCO, USGS, Food and Agriculture Organization, National
Park Service, Natural Resources Canada, GeoBase, IGN, Kadaster NL, Ordnance Survey,
METI, Esri Japan, Esri China, the GIS User Community, and © OpenStreetMap
contributors 2020. Distributed under a Creative Commons BY-SA License.


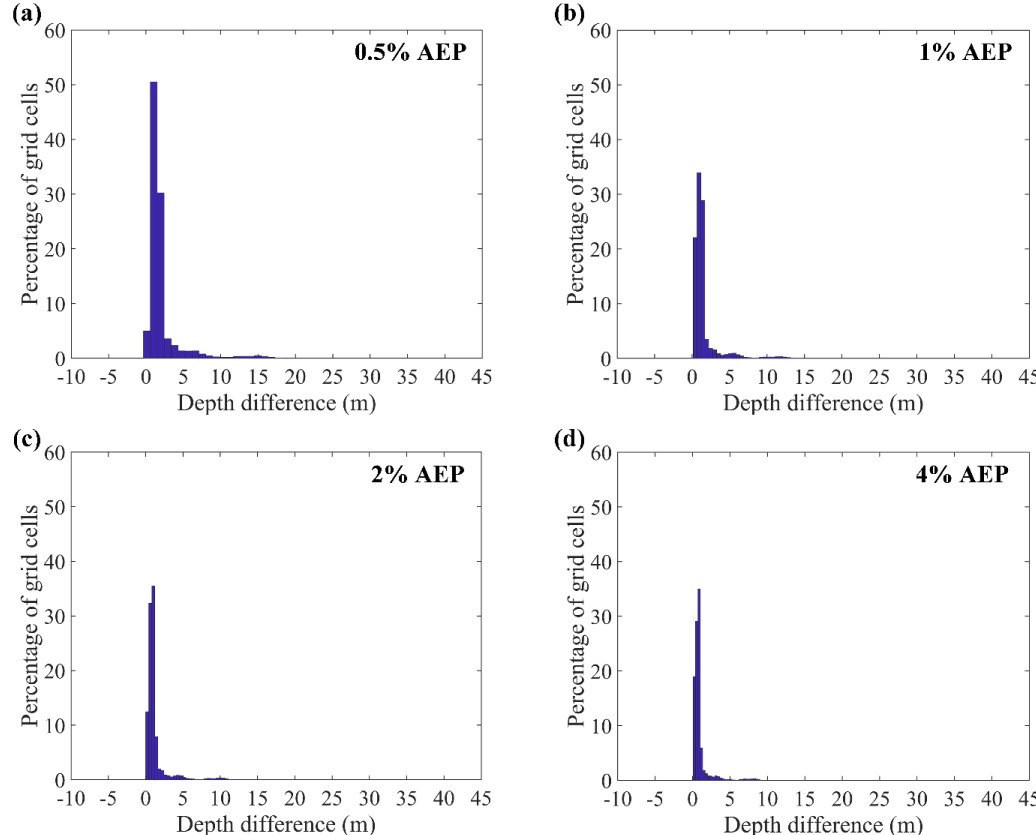

Figure 10. Histograms for the future changes (2011–2050) in the flood depth relative to the baseline period (1966–2005) for (a) 0.5%, (b) 1%, (c) 2%, and (d) 4% AEP flood depth frequency maps.

**3.6. Vulnerability of Electricity Infrastructure**

Figure 11a shows the box and whisker plot for the distributions of maximum flood depth values extracted at the substation location across all the baseline and future simulations, assuming that no flood protection measures were adopted (mitigation scenario 1). Of the 44 substations, 5 substations could have been affected during the





baseline period, while 7 substations are projected to be affected during the future period
(Figure 11a). Increases are indicated not only for the number of affected substations but
also for flood inundation depth values in the projected future climate. Overall, the mean
of the ensemble flood depth values shows an ~0.6 m increase in the future period (Figure
11a). Such an increase in the flood depth magnitude has the potential to exacerbate flood
related damage to electrical components, which can inflate the cost of hardening
measures such as elevating substations and constructing flood-protective barriers. As
expected, when the substations were flood-proofed up to BFE plus ~0.91 m (mitigation
scenario 2), the number of affected substations is reduced to three and four during the
baseline and future periods, respectively (Figure 11b). The locations of substations that
were impacted in the baseline period, in both mitigation scenarios, are consistent with the
Whitfield County Emergency Management Agency report map (EMA, 2016) that shows
the locations of critical facilities vulnerable to the historical flooding.

The maximum inundation durations at the affected substations are summarized in

Figure 12a (mitigation scenario 1) and Figure 12b (mitigation scenario 2). For both
mitigation scenarios and all affected substations, ensemble mean inundation durations
exhibited an increase under future climate condition. This increase in inundation duration
probably would render substations out of service for longer periods of time by making it
difficult to repair damaged substation equipment and restore grid services to customers.
The potential hazards and consequences may also extend to critical facilities that are
supplied by the affected substations. Similar to results presented in the previous sections,
these results demonstrate the need for improving existing flood mitigation measures by
incorporating the trends and uncertainties that originate from climate change. The
vulnerability analysis approach presented in this study will better equip floodplain
managers to identify the most vulnerable substations and to recommend suitable
adaptation measures, while allocating resources efficiently.

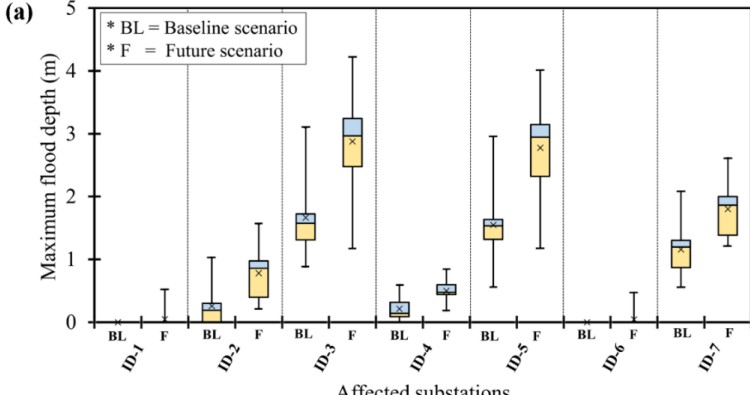

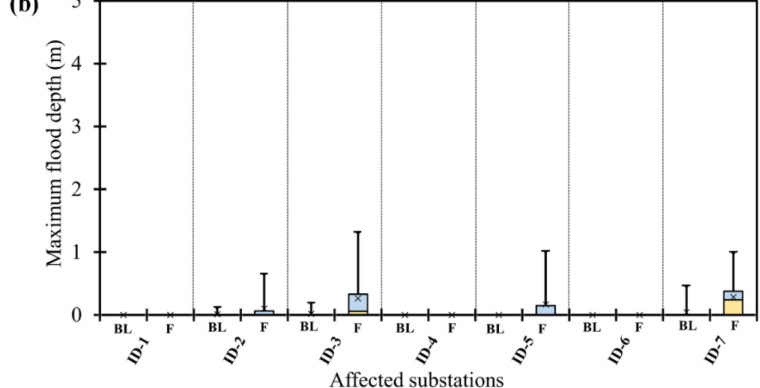


Figure 11. A summary of maximum flood depths for substations that were affected in the
baseline and/or future periods (a) without flood protection measures and (b) with flood
protection measures. Note: Affected substations with their corresponding IDs are shown
in Figure 1. There are no negative values in the vertical axis, as the minimum flood depth
value is zero.




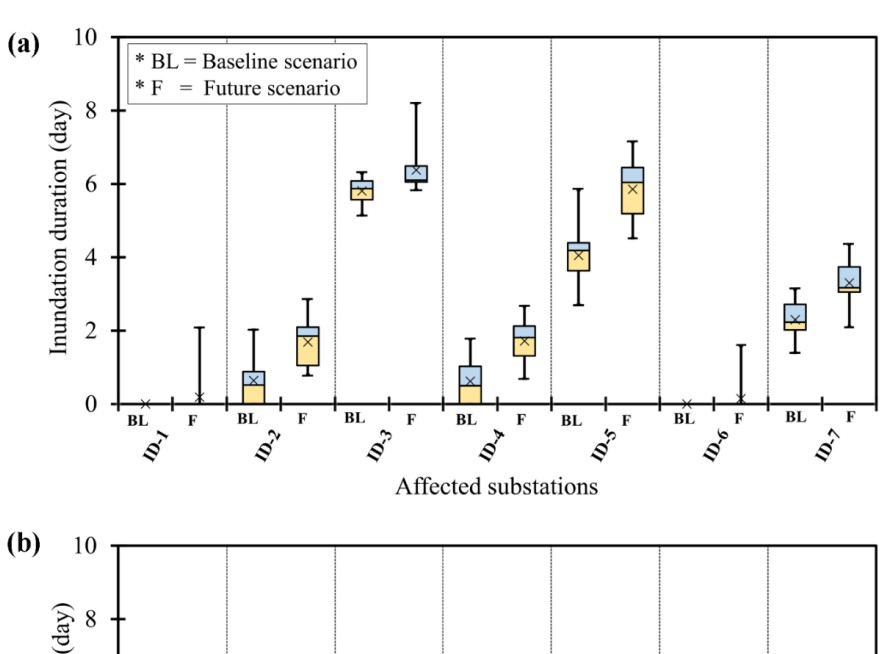


Figure 12. A summary of maximum inundation durations for substations that were

affected in the baseline and/or future periods (a) without flood protection measures and

(b) with flood protection measures. Note: Affected substations with their corresponding

IDs are shown in Figure 1. There are no negative values in the vertical axis, as the

minimum inundation duration is zero.



### 4. Summary and Conclusion

This paper applies an integrated modeling framework to evaluate climate change
impacts on flood regime, floodplain protection standards, and electricity infrastructures
across the Conasauga River Watershed in the southeastern United States. Our evaluation
is based on a climate-hydrologic-hydraulic modeling framework, which makes use of an
eleven member ensemble of downscaled climate simulations. Nine out of eleven
ensemble members project an increase in the flood inundation area in the future period.
Similarly, at the 1% AEP level, the flood inundation frequency curves indicate ~16 km$^2$
increase in floodplain area under the future climate. The comparison between the flood
depth frequency maps from the baseline and future simulations indicated that, on average,
~80% of grid cells exhibit a 0.2 to 1.5 m increase in the flood depth values. Without the
flood protection measures, of the 44 electric substations inside the watershed, 5 and 7
substations could be affected during the baseline and future periods, respectively. Even
after flood-proofing, three and four substations could still be affected in the baseline and
future periods. The increases in flood depth magnitude and inundation duration at the
affected substations in the future period will most likely damage more electrical
components, inflate the cost of hardening measures and render substations out of service
for a longer period of time.

Although future climate conditions are uncertain, our results demonstrate the needs
for (1) consideration of climate change in the floodplain management regulations; (2)
improvements in the conventional deterministic flood delineation approach through the
inclusion of probabilistic or ensemble-based methods, and (3) improvements in the
existing flood protection measures for critical electricity infrastructures through enhanced



hydro-meteorologic modeling capacities. In particular, rapidly advanced high-
performance computing capabilities have enabled the incorporation of computationally
intensive 2D hydraulics modeling in the ensemble-based hydroclimate impact
assessment. While the computational cost demonstrated in this study may still seem
steep, in the current speed of technology advancement, we will soon be able to implement
such a computationally intensive assessment for wide applications. The approach
presented in this study can be used by floodplain managers to develop flood depth
frequency maps and to identify the most vulnerable electric substations.
**Author Contribution**
*Dullo*, *Kalyanapu*, *Kao*, *Gangrade* and *Morales-Hernández* developed the concept for the
paper, designed the methodology and *Dullo* performed all the simulations required for the
study with feedback from all the co-authors. *Sharif*, *Ghafoor* and *Morales-Hernández*
focused on programming, software development and testing of existing code components.
*Ashfaq* and *Morales-Hernández* provided access to supercomputing machine hours on
ORNL's SUMMIT and RHEA computers. The manuscript was edited by *Dullo* with inputs
from the co-authors.
**Competing Interests**
The authors declare that they have no conflict of interest.
**Acknowledgments**

This study was supported by the US Air Force Numerical Weather Modeling

Program. TTD, MBS, AJK, and SG also acknowledge support by the Center of
Management, Utilization, and Protection of Water Resources at Tennessee Technological
University. Some portion of the project was funded by the UT Battelle Subcontract No:



4000164401. The research used resources of the Oak Ridge Leadership Computing
Facility at Oak Ridge National Laboratory. Some of the co-authors are employees of UT-
Battelle LLC under contract DE-AC05-00OR22725 with the US Department of Energy.
Accordingly, the US government retains and the publisher, by accepting the article for
publication, acknowledges that the US government retains a nonexclusive, paid-up,
irrevocable, worldwide license to publish or reproduce the published form of this
manuscript, or allow others to do so, for US government purposes. The input data sets are
cited throughout the paper, as appropriate.
**Data Availability**
The data that support the findings of this study are openly available in figshare
repository at the following URL:
https://figshare.com/projects/Conasauga_Flood_Modeling_Project/80840.

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
