# Peer review of "Conasauga River Watershed: An Application of Ensemble"

_Natural Hazards and Earth System Sciences, 2020_

## Referee Comment (RC1) · Anonymous Referee #1 · 2 Dec 2020

The manuscript by Dullo et al. titled, "Assessing Climate Change-Induced Flood Risk in the Conasauga River Watershed: An Application of Ensemble Hydrodynamic Inundation Modeling" presents a systematic approach for evaluating the impact of climate change on exacerbating the future flood risk across a large watershed. First, a hydrologic model is used to simulate streamflow corresponding to multiple climate projections and second, a high-resolution hydrodynamic model (TRITON) is used to simulate the flood inundation extents corresponding to the streamflow values for different scenarios. I appreciate how thoroughly the modeling is conducted and described in the

text. I particularly like the authors' approach to quantify flood frequency estimates at a grid-level. Overall, this is a strong paper but requires additional discussion and justification. Please see below for comments that are intended to improve the quality of the manuscript:

1. My major concern is the absence of a variable roughness distribution based on different land use types for current and future periods. The hydrodynamic model assumes a fixed channel and floodplain roughness which may not be reflective of future land use variability. Therefore, the study evaluates climate variability from a hydrologic perspective, but only uses the modified streamflow to drive the same hydrodynamic model. Climate change is strongly linked to human-induced land use change and therefore, the land use variability must be reflective in the future simulations. Similarly, channel roughness can also vary spatial from upstream to downstream in large channels. Please comment on why this is not incorporated and how this might influence results.

2. Is the initial depth modification a proxy for antecedent conditions? How would the results change if depth variability in Manning's n is considered? Usually, the channel and floodplain roughness reduce with increasing depth following an exponential function. This has been applied previously in GSSHA and ICPR (https://doi.org/10.1029/2019WR025769). Please comment on how the results might be impacted having not incorporated a depth-variable roughness distribution.

3. I know the LP3 distribution works well for streamflow, but I am not sure of its applicability for flood depths. I would assume using a log-normal distribution for curve fitting flood depths would be more optimal. Can the authors provide a comparative analysis of the two distributions? Did the authors consider different distributions for curve fitting? Please comment.

4. Lines 420-424: This result resembles what has been reported in Dey et al. 2019 (https://doi.org/10.1016/j.jhydrol.2019.05.085). Please add a statement highlighting this similar finding. Additionally, this study also highlights the impact of incorporating

an optimal channel shape. In the manuscript, the authors have modified the channel bottom, but this may not be entirely reflective of the bathymetric configuration of the streams. While channel shape and sinuosity may not impact 1D models where channel conveyance volumes are more important, this may be essential in 2D models. Please discuss the potential limitations of the approach adopted in this study.

---

## Referee Comment (RC2) · Anonymous Referee #2 · 5 Jan 2021

**Review to Dullo et al. "Assessing Climate Change-Induced Flood Risk in the Conasauga River Watershed: An Application of Ensemble Hydrodynamic Inundation Modeling"**

The manuscript (MS) presents a modeling approach for assessing the potential future impacts of climate change on the future flood risk in a watershed, according to greenhouse gas emission scenarios (RCPs).

The approach is based on a modeling chain combining an hydrological model to estimate flood hydrographs and a 2D hydrodynamic model to simulate flood inundation.

The MS is well prepared, and I appreciated the sensitivity analyses for the most important parameters of the hydrodynamic model. Also, other specific choices seem sufficiently justified. I would have given minor revisions for this MS, but I rather suggest major revisions as I think that the advancement respect to previous work Gangrade et al. (2019) should be better highlighted. I have read the comments of referee 1, which mainly focus on some limitations in the hydrodynamic modeling. I agree with most of them. Hence my comments will mainly focus on other aspects of the MS.

**Specific comments**

- Introduction and Conclusions and summary: Please better highlight the advancements respect to previous work by Gangrade et al. (2019), Journal of Hydrology https://doi.org/10.1016/j.jhydrol.2019.06.027

- L 365 referring to Fig. 2: the control and baseline samples of annual maximum peak streamflow (box-plots) may be seen as "significantly different" rather than "comparable". Indeed, two points need to be clarified in this respect: a) the shown baseline sample is relative to bias-corrected data or not? b) control and baseline samples have different lenghts, so, perhaps a more objective way of comparing them may be to apply some bootstrapping algorithm, or to randomly extract from the baseline sample several sub-samples having the same lenght of the control sample, and compare these somehow.

- Fig. 8. It may be possible to derive the analogous curve for the control scenario hydrographs. How does this compare to the shown baseline and future curves?

**Minor points**

- L 188: Many researchers consider as a standard choice a period of 30 years instead of 40 years. A comment on this may be added to the MS

- L 476: there is only an indirect demonstration that the model can reproduce well flow velocity. As no direct comparison is performed (data are not available in this sense, as far as I have understood), perhaps this should be downplayed.

- L298: A minimum threshold of 10 cm flood depth was used to judge whether a cell was dry or wet. How much do you think your results can be sensitive respect to this theshold value?

---

## Author Comment (AC1) · 4 Mar 2021

The authors would like to thank the reviewer for the insightful and constructive comments. We have reviewed the comments and provided our responses herein. The reviewer's comments are presented first followed by our response.

**Anonymous Referee #1**

The manuscript by Dullo et al. titled, "Assessing Climate Change-Induced Flood Risk in the Conasauga River Watershed: An Application of Ensemble Hydrodynamic Inundation Modeling" presents a systematic approach for evaluating the impact of climate change on exacerbating the future flood risk across a large watershed. First, a hydrologic model is used to simulate streamflow corresponding to multiple climate projections and second, a high-resolution hydrodynamic model (TRITON) is used to simulate the flood inundation extents corresponding to the streamflow values for different scenarios. I appreciate how thoroughly the modeling is conducted and described in the text. I particularly like the authors' approach to quantify flood frequency estimates at a grid-level. Overall, this is a strong paper but requires additional discussion and justification. Please see below for comments that are intended to improve the quality of the manuscript:

**R1.1.** My major concern is the absence of a variable roughness distribution based on different land use types for current and future periods. The hydrodynamic model assumes a fixed channel and floodplain roughness which may not be reflective of future land use variability. Therefore, the study evaluates climate variability from a hydrologic perspective, but only uses the modified streamflow to drive the same hydrodynamic model. Climate change is strongly linked to human-induced land use change and therefore, the land use variability must be reflective in the future simulations. Similarly, channel roughness can also vary spatial from upstream to downstream in large channels. Please comment on why this is not incorporated and how this might influence results.

**Our response:**

Thank you for the comment. The surface roughness values were selected based on the Whitfield County Flood Insurance Study (FIS, 2007; reference listed in the manuscript), which reported a range of main channel and floodplain Manning's n values. This is discussed under sensitivity analysis section (section 3.2). Further, to obtain representative roughness values, a couple of flood simulations were performed by adjusting the Manning's n values within the main channel and floodplain until a satisfactory agreement was achieved between the simulated and reference FEMA flood map. This is discussed in detail under flood model evaluation section (section 3.3).

We understand the reviewer's concern about the absence of variable roughness values based on land use in current and future periods. To model future land use and the corresponding surface roughness, one may use land use forecasts such as USGS's FOREcasting SCEnarios of Land-use Change (FORE-SCE) model for the Contiguous United States (CONUS) developed by Sohl et al. (2018). However, this dataset is only available at 250 m spatial resolution. Resampling this dataset to a 10 m spatial resolution will likely introduce more interpolation errors and may not adequately represent the spatial variability of land use patterns. This in turn will add additional uncertainty and hence requires an even more comprehensive task to characterize its impact.

The main focus of this manuscript is to evaluate the impacts of climate change on flood inundation extent and electricity infrastructures. Incorporating additional factors such as land use land cover change (LULCC) would increase the dimension of scenarios and require expanded ensemble simulations. These would require more computing resources and creates difficulty in data management as the total number of outputs increase significantly. Although we were unable to incorporate the suggested change in this study, the reviewer's comment is very essential in enhancing the analysis and model accuracy. As such, we have included additional information in section 3.3 to discuss the limitations of our current inundation modeling approach, such as missing variable Manning's n values and simplified river bathymetry correction. These limitations are provided as references for the enhancement of inundation modeling in future applications.

**R1.2.** Is the initial depth modification a proxy for antecedent conditions? How would the results change if depth variability in Manning's n is considered? Usually, the channel and floodplain roughness reduce with increasing depth following an exponential function. This has been applied previously in GSSHA and ICPR (https://doi.org/10.1029/2019WR025769). Please comment on how the results might be impacted having not incorporated a depth-variable roughness distribution.

**Our response:**

The initial water depth values represent the starting water surface elevation along the main channel (water course).

Dynamic variability of Manning's n value was not a part of the current study, as our TRITON model does not simulate this phenomenon at this time. However, as the reviewer suggested, if depth variability in Manning's n could be considered, it is likely that the channel and floodplain roughness would change and likely increase in part due to incorporating additional flow turbulence during bankfull flows (Morvan et al., 2008; Christelis et al., 2016; Bellos et al., 2018), as well as additional losses from complex floodplain flows that occurs during high flow events. However, the authors are hesitant to comment on the results without providing any evidence as this would lead to speculation but not likely affect the scope and outcome of the current study. We have discussed this in the limitations of our current inundation modeling approach and also included the suggested Saksena et al. (2019) reference in the revised manuscript.

While it may be worthy to investigate the potential impacts by considering depth variability in Manning's n, it is unlikely to impact the main objective of the study which is to demonstrate the applicability of a computationally intensive ensemble inundation approach to study the climate change impacts on flood regimes, floodplain regulation standards, and the vulnerability of existing infrastructures. The point by the reviewer is well taken and will be considered in future studies.

**R1.3.** I know the LP3 distribution works well for streamflow, but I am not sure of its applicability for flood depths. I would assume using a log-normal distribution for curve fitting flood depths would be more optimal. Can the authors provide a comparative analysis of the two distributions? Did the authors consider different distributions for curve fitting? Please comment.

**Our response:**

Thank you for the insightful comment. Indeed, although the Log-Pearson type III (LP3) distribution was recommended by Bulletin 17C (England et al., 2019) for streamflow, it may not be the optimal choice for flood depth. However, given the community's familiarity with LP3, we still decided to test the applicability of LP3 in this study. Our goodness-of-fit tests suggested that LP3 can still be a reasonable choice for flood depths.

Based on the reviewer's suggestions, we have conducted an evaluation by randomly selecting 679 locations in the study area and comparing the fittings of both LP3 and Log-Normal (LN) distributions. These locations were identified by sampling at 500 m interval along the streams in the domain, which resulted in a total of 851 points. Out of the 440 simulations, if any of these points were not wet (i.e., the depth is not greater than 10 cm) for 30 or more simulations, they were excluded from further analysis. This resulted in 679 points within the computational domain for the next step.

Using the Anderson-Darling (AD) goodness-of-fit test ($\alpha = 0.05$), we found that LP3 is accepted in more locations than LN. Additionally, we also used Akaike information criterion (AIC) to evaluate the suitability of both distributions and we found that LP3 can outperform LN in more locations. The results indicated that the LP3 can be an even more suitable choice than LN, for our study area (Table R1).

*Table R1 – AD and AIC comparison between LP3 and LN distributions at 679 locations*

|  | Log-Normal | Log-Pearson Type III |
|---|---|---|
| AD p-value > .05 | 590 | 636 |
| AD p-value ≤ .05 | 89 | 43 |
| Suitability based on AIC | 235 locations | 444 locations |

It must be noted, however, that our goal in this study is not to identify the most suitable choice of distribution for flood depth. Therefore, there can be other more suitable distributions than the two tested herein. Given the good performance of LP3, we believe that it's sufficient for the purpose

of our study. These additional analysis and clarification have been included in Section 3.5 of the revised manuscript.

The distribution fitting (LP3 and LN), Anderson-Darling k-sample test and AIC calculations were conducted using Python 3 and SciPy libraries (Hovey & DeFiore, 2003; Salvosa, 1930; Scholz & Stephens, 1987; Virtanen et al., 2020; Vogel & McMartin, 1991).

**R1.4.** Lines 420-424: This result resembles what has been reported in Dey et al. 2019 (https://doi.org/10.1016/j.jhydrol.2019.05.085). Please add a statement highlighting this similar finding. Additionally, this study also highlights the impact of incorporating an optimal channel shape. In the manuscript, the authors have modified the channel bottom, but this may not be entirely reflective of the bathymetric configuration of the streams. While channel shape and sinuosity may not impact 1D models where channel conveyance volumes are more important, this may be essential in 2D models. Please discuss the potential limitations of the approach adopted in this study.

**Our response:**

Thank you for the comment and suggested reference. We have revised the manuscript to highlight the similar findings from other studies. Further, we have included statements such as "*Although this simple approach is unlikely to adjust the channel bathymetry to its true values, it can improve the channel conveyance volume that is lost in the base DEM.*" in section 3.3 to discuss our model limitations. The suggested Dey et al. (2019) reference has been included.

**References:**

Bellos, V., Nalbantis, I., and Tsakiris, G. (2018). Friction Modeling of Flood Flow Simulations, Journal of Hydraulic Engineering, 144(12), DOI: 10.1061/(ASCE)HY.1943-7900.0001540

Christelis, V., Bellos, V. and Tsakiris, G. (2016). "Employing surrogate modelling for the calibration of a 2D flood simulation model." In Proc., 4th European Congress of IAHR, "Sustainable Hydraulics in the Era of Global Change", edited by S. Erpicum, B. Dewals, P. Archambeau, and M. Pirotton, 727-732, Liege, Belgium: CRC Press.

Dey, S., Saksena, S., and Merwade, V.: Assessing the effect of different bathymetric models on hydraulic simulation of rivers in data sparse regions, J. Hydrol., 575, 838-851, doi:10.1016/j.jhydrol.2019.05.085, 2019.England, J.F., J., Cohn, T. A., Faber, B. A., Stedinger, J. R., Thomas, W.O., J., Veilleux, A. G., Kiang, J. E., & Mason, R.R., J. (2019). Guidelines for Determining Flood Flow Frequency Bulletin 17C Book 4, Hydrologic Analysis and Interpretation. In Book: (Issue May). https://pubs.usgs.gov/tm/04/b05/tm4b5.pdf

Hovey, P., & DeFiore, T. (2003). Using Modern Computing Tools to Fit the Pearson Type III Distribution to Aviation Loads Data. http://www.tc.faa.gov/its/worldpac/techrpt/ar03-62.pdf

Morvan, H., Knight, D., Wright, N., Tang, X., and Crossley, A. (2008). The concept of roughness in fluvial hydraulics and its formulation in 1D, 2D and 3D numerical simulation models, Journal of Hydraulic Research, 46:2, 191-208, DOI: 10.1080/00221686.2008.9521855

Salvosa, L. R. (1930). Tables of Pearson's Type III Function. The Annals of Mathematical Statistics, 1(2), 191–198. https://doi.org/10.1214/aoms/1177733130

Scholz, F. W., & Stephens, M. A. (1987). K-sample Anderson–Darling tests. Journal of the American Statistical Association, 82(399), 918–924. https://doi.org/10.1080/01621459.1987.10478517

Sohl, T.L., Sayler, K.L., Bouchard, M.A., Reker, R.R., Freisz, A.M., Bennett, S.L., Sleeter, B.M., Sleeter, R.R., Wilson, T., Soulard, C., Knuppe, M., and Van Hofwegen, T., 2018, Conterminous United States Land Cover Projections - 1992 to 2100: U.S. Geological Survey data release, https://doi.org/10.5066/P95AK9HP.

Virtanen, P., Gommers, R., Oliphant, T. E., Haberland, M., Reddy, T., Cournapeau, D., Burovski, E., Peterson, P., Weckesser, W., Bright, J., van der Walt, S. J., Brett, M., Wilson, J., Millman, K. J., Mayorov, N., Nelson, A. R. J., Jones, E., Kern, R., Larson, E., … Vázquez-Baeza, Y. (2020). SciPy 1.0: fundamental algorithms for scientific computing in Python. Nature Methods, 17(3), 261–272. https://doi.org/10.1038/s41592-019-0686-2

Vogel, R. W., & McMartin, D. E. (1991). Probability Plot Goodness-of-Fit and Skewness Estimation Procedures for the Pearson Type 3 Distribution. Water Resources Research, 27(12), 3149–3158. https://doi.org/10.1029/91WR02116

---

## Author Comment (AC2) · 4 Mar 2021

The authors would like to thank the reviewer for the insightful and constructive comments. We have reviewed the comments and provided our responses herein. The reviewer's comments are presented in first followed by our response.

**Anonymous Referee #2**

**R2.1.** Introduction and Conclusions and summary: Please better highlight the advancements respect to previous work by Gangrade et al. (2019), Journal of Hydrology https://doi.org/10.1016/j.jhydrol.2019.06.027

**Our response:**

The main difference between Gangrade et al. (2019) and this study is on the type of flood events. Gangrade et al. (2019) focused on evaluating flood risks associated with probable maximum flood events (AEP < $10^{-4}$) that are considered as the physics-based upper bound of surface inundation. In this study, we focused on more frequent extreme streamflow events (i.e., AEP around 1–0.2%) that are involved in broader engineering applications. Given their distinct nature and AEPs, different analyses and modeling strategies are hence needed. Also, the study area is different between these two studies. We have clarified the differences between Gangrade et al. (2019) and current study in various locations of this revised manuscript.

**R2.2.** L 365 referring to Fig. 2: the control and baseline samples of annual maximum peak streamflow (box-plots) may be seen as "significantly different" rather than "comparable". Indeed, two points need to be clarified in this respect: a) the shown baseline sample is relative to bias-corrected data or not? b) control and baseline samples have different lengths, so, perhaps a more objective way of comparing them may be to apply some bootstrapping algorithm, or to randomly extract from the baseline sample several sub-samples having the same length of the control sample, and compare these somehow.

**Our response:**

Thank you for the insightful and constructive suggestion. In addition to the box plot comparison, we have also conducted a two-sample t-test (α = 0.05) to compare if the means of control and baseline annual maximum streamflow are statistically significantly different. The two-tailed analysis resulted in a p-value of 0.093 (see Table R1), which suggests that there is no significant difference between the means and support the statement that the control and baseline annual maximum streamflow values are comparable. To address this concern, the following statement has been included in the manuscript (Line 374): "*In addition, we also conducted a two-tailed two-sample t-test (α = 0.05) to compare if the means of control and baseline annual maximum*

*streamflow are statistically different. The results yielded a p-value of 0.09 which suggested that there is no significant difference between the means of both control and baseline simulations.*"

*Table R1 - t-Test: Two-Sample Assuming Unequal Variances*

|  | Control | Baseline |
| --- | --- | --- |
| Mean | 331.92 | 388.23 |
| Variance | 30955.45 | 26239.07 |
| Observations (n)* | 32 | 440 |
| Hypothesized Mean Difference | 0 | |
| df | 34 | |
| t Stat | -1.73 | |
| t Critical two-tail | 2.032 | |
| P two-tail | 0.093 | |

* (n-1) degrees of freedom used for standard deviation calculation

**R2.3.** Fig. 8. It may be possible to derive the analogous curve for the control scenario hydrographs. How does this compare to the shown baseline and future curves?

**Our response:**

Although numerically speaking one may derive a similar curve only based on 32 years of control simulation results, it can be misleading and biased given the much smaller sample size. The baseline and future curves in Fig. 8 were each derived from 440 years of data to reduce uncertainty and render more confidence in the evaluation of lower AEP estimates. However, the control scenario consists of only 32 years of data which may not be suitable to support a meaningful comparison in Fig. 8.

**R2.4.** L 188: Many researchers consider as a standard choice a period of 30 years instead of 40 years. A comment on this may be added to the MS.

**Our response:**

Based on our understanding, a minimum of 30-year period was used in many studies (e.g., Alfieri et al., 2015a, 2015b) so that one may have a sufficiently long temporal window to capture the multi-decadal climate variability. Given the additional data provided by Gangrade et al. (2020), we have adopted a longer 40-year period which may further enlarge the sample space to better support the statistical analyses in this study. This additional clarification has been included in the revised manuscript.

**R2.5.** L 476: there is only an indirect demonstration that the model can reproduce well flow velocity. As no direct comparison is performed (data are not available in this sense, as far as I have understood), perhaps this should be downplayed.

**Our response:**

Thank you for the suggestion. The authors agree that the accuracy of simulated velocities is not evaluated within the scope of this study. To avoid overstatement, the manuscript has been modified by omitting "velocities" in the original statement. The statement now reads *"...demonstrating that the TRITON could reasonably estimate flood inundation extent and depths in the CRW."*

**R2.6.** L298: A minimum threshold of 10 cm flood depth was used to judge whether a cell was dry or wet. How much do you think your results can be sensitive respect to this threshold value?

**Our response:**

Thank you for your comment. With regards to the threshold value, Gangrade et al. (2019) tested the sensitivity of the threshold value in their study. Aside from the minimum threshold of 10cm, they also tested a minimum threshold of 1cm and reported minimal impact in maximum inundation area. Based on this fact, we can reasonably conclude that the results are marginally sensitive to the threshold values lower than 10cm. It is also important to note that this threshold value only applies to the analysis of post-processing results, as the accuracy with which TRITON decides whether a cell is dry or wet during the calculation is $10^{-12}$ meters.

**References:**

Gangrade, S., Kao, S. C., Dullo, T. T., Kalyanapu, A. J., & Preston, B. L. (2019). Ensemble-based flood vulnerability assessment for probable maximum flood in a changing environment. *Journal of Hydrology*, *576*, 342–355. https://doi.org/10.1016/j.jhydrol.2019.06.027

Gangrade, S., Kao, S. C., & McManamay, R. A. (2020). Multi-model Hydroclimate Projections for the Alabama-Coosa-Tallapoosa River Basin in the Southeastern United States. *Scientific Reports*, *10*(1), 2870. https://doi.org/10.1038/s41598-020-59806-6